# ROSA: Harnessing Robot States for Vision-Language and Action Alignment

## Abstract

Vision-Language-Action (VLA) models have recently made significant advance in multi-task, end-to-end robotic control, due to the strong generalization capabilities of Vision-Language Models (VLMs). A fundamental challenge in developing such models is effectively aligning the vision-language space with the robotic action space. Existing approaches typically rely on directly fine-tuning VLMs using expert demonstrations. However, this strategy suffers from a spatio-temporal gap, resulting in considerable data inefficiency and heavy reliance on human labor. Spatially, VLMs operate within a high-level semantic space, whereas robotic actions are grounded in low-level 3D physical space; temporally, VLMs primarily interpret the present, while VLA models anticipate future actions. To overcome these challenges, we propose a novel training paradigm, ROSA, which leverages robot state estimation to improve alignment between vision-language and action spaces. By integrating robot state estimation data obtained via an automated process, ROSA enables the VLA model to gain enhanced spatial understanding and self-awareness, thereby boosting performance and generalization. Extensive experiments in both simulated and real-world environments demonstrate the effectiveness of ROSA, particularly in low-data regimes.

## 1 Introduction

In recent years, Vision-Language Models (VLMs) (Liu et al., 2023; Karamcheti et al., 2024; Alayrac et al., 2022) have achieved remarkable progress in visual-linguistic understanding, demonstrating strong performance and generalization capabilities. Building on this success, there is growing interest in extending VLMs to end-to-end robotic control by developing generalist robot policies—commonly referred to as Vision-Language-Action (VLA) models (Kim et al., 2024; Brohan et al., 2023; Black et al., 2024). A key challenge in this direction is aligning the vision-language representation space with the robotic action space. To address this, a widely adopted approach (Brohan et al., 2023) is to directly fine-tune pretrained VLMs using large-scale expert action data, mapping visual observations and language instructions to the corresponding actions.

However, this direct fine-tuning paradigm suffers from significant limitations due to the spatial and temporal domain gaps inherent in the alignment process, leading to substantial data inefficiency. As shown on the left of Fig. 1, VLMs are typically pretrained on large-scale visual question answering datasets, where their feature representations primarily capture high-level semantics—e.g., identifying an object as a "banana." In contrast, robotic control, as illustrated on the right of Fig. 1, requires fine-grained spatial reasoning. For instance, beyond recognizing a banana, the model must accurately infer its 3D position to enable successful grasping. This mismatch between high-level semantic understanding and the need for precise spatial localization presents a significant challenge in aligning VLMs with robotic tasks.

Furthermore, while VLMs excel at interpreting the current semantic content of an image, VLAs must reason over time to forecast and plan future robotic actions. This introduces a current-to-future temporal gap, further complicating the alignment process. Combined with the spatial gap, these challenges necessitate large quantities of expert action data to effectively bridge the discrepancy during VLM fine-tuning. Moreover, this heavy data requirement significantly increases the burden of human data collection, impeding the rapid development of VLAs. In scenarios with limited expert data, the risk of overfitting becomes more pronounced, potentially degrading generalization perfor-

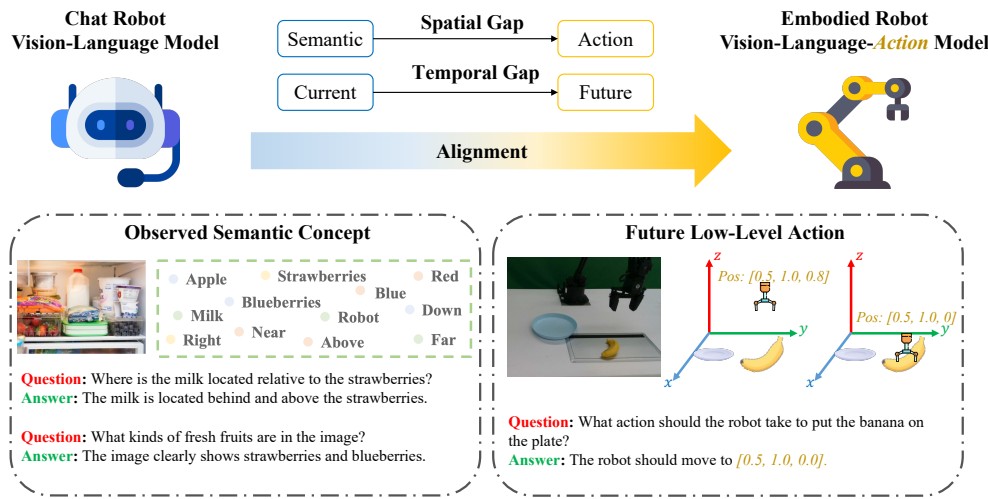

Figure 1: The spatial and temporal gaps in adapting VLMs to VLAs. VLMs are pretrained with large-scale VQA datasets to observe current high-level semantics in images, while VLAs are designed to predict low-level future actions in 3D space. The spatial-temporal gap poses challenges to the alignment process and results in data inefficiency in developing VLAs.

mance and restricting the model's applicability to novel tasks or environments. Recent works (Niu et al.; Zheng et al., 2024; Li et al., 2025b) have made initial attempts to alleviate the fine-tuning gap and reduce data requirements. Some (Li et al., 2024c; Niu et al.; Zheng et al., 2024) introduce additional supervision signals via extra annotation tools. Others (Wen et al., 2025; Li et al., 2025b) reduce data dependency by adopting more lightweight networks. However, these approaches either compromise model capacity or incur additional computational overhead from annotation tools, limiting their scalability.

In this paper, to address the aforementioned spatial and temporal gaps, we propose a novel training paradigm, **ROSA—RO**bot **S**tate estimation for vision-language and action **A**lignment. ROSA decomposes the alignment process into two complementary components: one dedicated to estimating the robot's current state, and the other focused on predicting future actions. Concretely, in addition to using standard expert action data for future action prediction, we introduce a novel form of robot state estimation data, which supervises the model to infer the robot's current state from the given image. The robot state includes the 3D position and orientation of the end-effector, as well as the gripper's open/closed status.

The robot state estimation task serves two key purposes. First, it explicitly enhances the model's ability to capture fine-grained spatial information. Second, it complements expert demonstrations by covering a broader portion of the action space, including regions underrepresented in expert data. This dual role helps bridge the spatial gap between pretrained VLMs and VLA models. Furthermore, by requiring the model to infer the robot's current state, ROSA provides a clearer and more structured spatial context, which in turn facilitates more accurate forecasting of future actions—thereby helping to close the temporal gap as well.

Collecting robot state data can be fully automated with virtually no additional human labor. Specifically, the robot performs plausible random actions via automated scripts within a predefined environment, during which observations and corresponding states are recorded. To enable joint training with expert data in VLA models, we structure the robot state estimation data to share the same format as expert demonstrations. This low-cost and easily scalable data acquisition approach makes ROSA a practical and scalable solution for aligning vision-language and action spaces, facilitating more data-efficient training of VLAs and ultimately improving performance.

We conduct extensive experiments to evaluate the effectiveness of ROSA. Using a standard VLA model, we perform controlled studies in both the RLBench simulation environment and on a real-world WidowX robot platform. Our results show that ROSA significantly enhances VLA performance and generalization ability. The improvement is especially significant in real-world low-data scenarios, where ROSA even doubles the success rate compared to the baseline.

To summarize, our contributions are as follows:

- We propose a novel training paradigm named ROSA that harnesses robot state estimation data to achieve better alignment between the vision-language and action spaces.

- We propose a simple yet effective solution to create the robot state estimation data that significantly enhances VLA's data efficiency without requiring additional human collection efforts.

- We conduct extensive experiments on both the RLBench simulation and a real-world WidowX platform, demonstrating that ROSA effectively enhances current VLA models and achieves superior performance compared to previous methods.

## 2 RELATED WORKS

### 2.1 VISION-LANGUAGE-ACTION MODELS.

Building a generalist policy (Mees et al., 2024; Shridhar et al., 2023; Brohan et al., 2022) has long been a central goal in robotic manipulation. In recent years, the impressive performance and generalization capabilities demonstrated by Vision-Language Models (VLMs) (Liu et al., 2023; Li et al., 2024a; Alayrac et al., 2022) have inspired researchers to develop robot policies based on VLMs, commonly referred to as Vision-Language-Action (VLA) models (Li et al., 2024b; Vuong et al., 2023; Zhou et al., 2025; Belkhale et al., 2024; Li et al., 2024d). These models have shown great promise in enabling robots to perform a wide range of tasks with enhanced generalization ability. Among them, RT-2 (Brohan et al., 2023) stands out as a typical pioneering work, which is jointly trained on web-scale VQA data and robotic demonstrations, showcasing impressive performance. Following it, OpenVLA (Kim et al., 2024)adopts a similar approach as one of the earliest open-source VLA models, trained on the large-scale OpenX-Embodiment dataset (Vuong et al., 2023). LLARVA (Niu et al.) collects trajectory annotations as auxiliary tasks to enhance the accuracy of action prediction. CogACT (Li et al., 2024b) introduces architectural modifications to generate continuous action chunks, employing a large diffusion action head.

### 2.2 DATA-EFFICIENT VISION-LANGUAGE-ACTION MODELS.

The high data demand of VLA models is a well-recognized challenge. Some prior works (Brohan et al., 2023; Vuong et al., 2023) aim to address this issue by incorporating large amount of expert demonstrations. For example, RT-X (Vuong et al., 2023) collects large-scale cross-platform expert trajectories with great human labor to train a general and high-performing VLA model. Some works (Zheng et al., 2024; Li et al., 2024c) introduce additional annotated data to help with the training. For instance, TraceVLA (Zheng et al., 2024) injects additional supervision by introducing external detectors to generate visual trajectory annotations. LLaRA (Li et al., 2024c) leverages meta annotations such as the bounding boxes to construct auxiliary spatial reasoning tasks, thereby improving model performance under low-data conditions. Another line of works (Wen et al., 2025; Li et al., 2025b; Zhang et al., 2025) reduce the model size to mitigate data dependency. For example, TinyVLA (Wen et al., 2025) focuses on building a fast VLA with one billion LLM backbone. In addition, works like (Li et al., 2025a) adopts atomic skill library construction to improve data efficiency by decomposing complex tasks into reusable primitive skills. Our approach is also data-efficient, but with a key difference: we do not rely on any additional human labor for collection or extra labeling modules, nor do we compromise model capacity.

### 2.3 ROBOT STATE ESTIMATION

Robot state estimation is a task in which, given input RGB images or other information such as depth, the model is required to estimating the position and orientation of a robot or its components in 3D space. This is a critical topic in both robotics and computer vision, such as autonomous navigation (Kothari et al., 2017; Prusak et al., 2008) and human-robot interaction (Christen et al., 2023; Svenstrup et al., 2009; Xia et al., 2020), with several representative works (Heindl et al., 2019; Miseikis et al., 2018; Xu et al., 2022; Ban et al., 2024). Our work draws inspiration from this task but with difference in the objectives and applications. Specifically, our model only needs

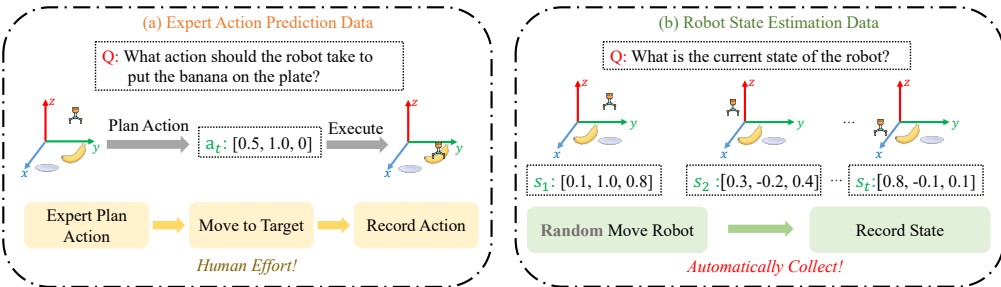

Figure 2: Illustration of the two types of data used by ROSA to train VLA models. (a). Expert action prediction data, which requires human effort to collect. (b). Robot state estimation data, which is obtained automatically without human collection by letting the robot move randomly.

to predict the pose of the end-effector and the gripper's open-close status that are necessary for robot manipulation tasks, without needing to estimate full-body configurations. Furthermore, the goal of our estimation is not for this task itself, but rather to serve as auxiliary supervision for our downstream robot control task, which enhances the spatial awareness of VLAs, improving VLAs' performance in action prediction.

## 3 METHOD

In this section, we present a comprehensive overview of our proposed ROSA. We first introduce the training data ROSA use in Sec. 3.1, especially the robot state estimation data. Then we provide the model architecture of ROSA in Sec. 3.2.

### 3.1 ROSA TRAINING DATA

The primary objective of ROSA is to effectively adapt a pretrained VLM to the robotic action space, enabling the model to directly generate control signals for manipulating robots based on visual observations and language instructions. To better align vision-language representations with robot actions, ROSA addresses the alignment problem through two complementary components: one for anticipating upcoming actions, and the other for accurately capturing the robot's current state. This decomposition explicitly encourages the model to develop a strong capability for 3D spatial understanding and accurate self-perception, both of which are essential foundations for effective action prediction. As illustrated in Fig. 2, ROSA leverages two distinct types of data including the expert action prediction data and the robot state estimation data to jointly fine-tune the pretrained VLM.

**Expert action data:** As shown in Fig. 2 (a), to obtain expert actions, human operators are required to carefully collect demonstrations by manually guiding the robot to target positions to complete different tasks. Formally, we assume an expert action dataset $\mathcal{D} = \{D_1, D_2, \ldots, D_m\}$ of m demonstrations across various tasks, where each demonstration $D_i = \{(o_1^i, a_1^i, l^i), (o_2^i, a_2^i, l^i), \ldots (o_t^i, a_t^i, l^i), \ldots\}$ contains a variable number of instruction-observation-action pairs. Here, $o_t^i, a_t^i, l^i$ denote the visual observation, expert action and language instruction for the $i$-th demonstration at timestep $t$, respectively. The action $a_t$ consists of a 7-DoF (degrees of freedom) control signal required to manipulate the robot's end-effector, including 3D position, orientation, and the gripper's open/close status:

$$a_t = [x, y, z, \phi, \theta, \psi, g], \tag{1}$$

where $(x, y, z)$ denotes the gripper's position, $(\phi, \theta, \psi)$ denotes the Euler angles, and $g \in \{0, 1\}$ denotes the opening status of the gripper (1 for open).

**Robot State Estimation Data:** While expert action data can directly supervise the VLA model to learn the action prediction objective, collecting such data is costly and labor-intensive. In contrast, we propose to collect the robot state estimation through a fully automated pipeline. This type of data can complement expert action data in training the VLA model without requiring additional human effort, reducing the model's reliance on large amounts of manually collected demonstrations.

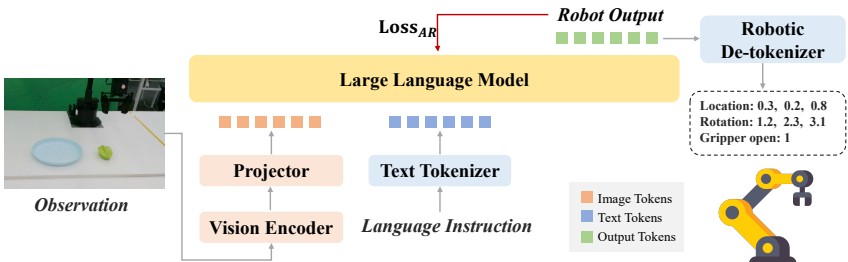

Figure 3: Overview of the ROSA architecture. ROSA adopts a classic VLM architecture. Image observations are encoded into image tokens by a vision encoder and a projector. These image tokens are combined with text tokens and fed into an LLM. The model is trained with an autoregressive next-token prediction objective.

Our robot state data collection pipeline proceeds as follows (Fig. 2 (b)). We first initialize a specific scene configuration—for example, placing a plate and a banana randomly on the table for a put-banana-into-plate configuration. Based on the scene setup, we choose a feasible action space to ensure that the robot's movements remain within safe bounds, avoiding collisions with objects in the environment that may lead to positioning errors. We then allow the robot to perform random movements within this constrained space and record robot's states at each time step. Concretely, we discretize the feasible 3D action space into uniform grids of resolution $n$, yielding $n^3$ candidate waypoints, from which we uniformly sample targets. For each sampled waypoint, the controller moves the robot's end-effector to the target pose, and we record the corresponding image at the target pose and the waypoint as the current state. By repeating this process, we collect diverse robot state data that uniformly covers the action space, forming a robot state estimation dataset.

To ensure that the robot state data can be effectively integrated with expert action data for joint training of the VLA model, we collect robot states that capture the same 7 degrees of freedom as the expert actions, including the end-effector's position, orientation, and gripper status. Therefore, in terms of format, the action and the state are identical. The key difference lies in their semantics: the action represents the target state that the robot should reach at the next time step, while the state means the robot states at current time step. The structural homogeneity between state data and expert action data ensures that the model is trained with a consistent target domain across both data types. Moreover, to construct a structurally consistent dataset, we pair each robot state sample with a uniform language instruction: *"What is the current state of the robot?"*, ensuring that the format mirrors that of the expert action data and enabling unified training under the VLA framework. Formally, we construct a robot state estimation dataset $\mathcal{S} = \{e_1, e_2, \ldots, e_k\}$ of $k$ pairs, where $e_t = (o_t, s_t, l_{\text{state}})$, and $o_t, s_t, l_{\text{state}}$ represent the observation, robot state at timestep $t$ and the state language instruction, respectively.

### 3.2 MODEL ARCHITECTURE

In this section, we detail the model architecture of ROSA, which is built upon standard LLaVA (Liu et al., 2023). We elaborate on three key components: the vision-language modules, robotic tokenization and de-tokenization, and the training objective.

**Vision-language Modules:** As illustrated in Fig. 3, the model takes two types of input: a language instruction $l$ that specifies the task the robot is expected to perform and a visual observation $o_t$ consisting of a single front view RGB image. The language instruction is encoded by a text encoder into a sequence of text tokens $Z_t$. Meanwhile, the visual observation is processed by a vision encoder $f_{\text{vis}}$ to extract visual features $H_v$. These visual features are then mapped into the same embedding space as the text tokens by a projector $f_{\text{proj}}$, resulting in visual tokens $Z_v$. The visual tokens and text tokens are then concatenated together and fed into a large language model $f_{\text{llm}}$, which performs causal reasoning over the input tokens and outputs a sequence of robot tokens $R$. The whole process can be formulated as follows:

$$R = f_{\text{llm}}([Z_v, Z_t]), \quad Z_v = f_{\text{proj}}(f_{\text{vis}}(o_t)). \tag{2}$$

**Robotic Tokenization and De-tokenization:** To allow the large language model to predict robot actions and states, we convert continuous robot actions and states into discrete values that serve as the LLM's output tokens. Take the robot's position along the x-axis as an example, given $x_i \in [x_{\min}, x_{\max}]$, we apply a linear quantization function to map it to an integer token $X_i \in \{0, 1, \ldots, \text{bin\_size} - 1\}$ as follows:

$$X_i = \left\lfloor \frac{(x_i - x_{\min})}{x_{\max} - x_{\min}} \times (\text{bin\_size} - 1) \right\rfloor \tag{3}$$

For instance, if bin\_size $= 256$, a possible action sequence will be '183 180 36 0 127 49 255'. During inference, we de-tokenize the predicted token to recover the continuous action or state values by performing the inverse mapping:

$$\hat{x}_i = x_{\min} + \frac{X_i}{\text{bin\_size} - 1} \times (x_{\max} - x_{\min}) \tag{4}$$

**Training Objective:** We jointly train the model using both expert action data $\mathcal{D}$ and robot state data $\mathcal{S}$. For both types of data, we employ a unified training objective: the next-token-prediction cross-entropy loss, defined as follows:

$$\mathcal{L} = -\sum_i \log P(y_i \mid y_{<i}, \mathbf{o}, \mathbf{l}; \omega) \tag{5}$$

where $y_i$ represents the $i$-th token and $\omega$ demotes the parameters of the model.

## 4 EXPERIMENT

### 4.1 EXPERIMENT SETUP

We evaluate ROSA in both the RLBench (James et al., 2020) simulation environment and on a real-world WidowX robot.

**RLBench**. We train and evaluate ROSA on 12 RLBench tasks, using a fixed front-facing RGB camera with a resolution of $336 \times 336$ to capture the visual input and a Franka Panda robot equipped with a parallel gripper. Following prior work (Shridhar et al., 2023), we evaluate performance over 25 episodes per task and report the average success rate (SR). Each evaluation is repeated three times to obtain the final score. More details are provided in the Appendix.

**Real-World WidowX Robot.** We use a WidowX 250S robot for real-world experiments. We evaluate on four seen tasks and four generalization tasks. The seen tasks include three short-range tasks—*Banana in Plate*, *Strawberry in Bowl*, *Starfruit in Plate*—and one long-range task: *Banana and Strawberry in Bowl*. The generalization tasks consist of *Cube in Plate* (unseen object), *Strawberry in Box* (unseen container), *Cube in Box* (unseen object and container), and *Strawberry in Bowl* (with distractors). Each task is evaluated over 10 trials. More details are described in the Appendix.

### 4.2 QUANTITATIVE RESULTS

#### 4.2.1 EFFECTIVENESS OF ROSA

We conduct experiments to show the performance of ROSA on RLBench and the real-robot across different data scales of expert action data. For simulation evaluation, as shown in Tab. 1, given the same amount of expert action data, ROSA consistently outperforms the baseline model at all scales, with particularly large gains when expert data is limited. Notably, with only 50 or 100 expert demonstrations, ROSA achieves a 7.1% and 11.4% improvement in average SR on RLBench, respectively. On real robot, as show in Fig. 4, the effectiveness is more pronounced with a 35% average SR improvement on the real robot. We attribute this to the greater variability present in real-world environments, such as cluttered backgrounds and lighting changes. By leveraging diverse state samples, ROSA enhances robustness under such challenging conditions.

**Sufficient Data Scenarios.** As shown in Tab. 1, the performance of VLA models consistently improves with increasing data scale. Nevertheless, even when the amount of expert action data is sufficiently large (i.e., 500 episodes per task), ROSA still achieves a 1.6% improvement in average

Figure 4: Comparison of performance between ROSA and the baseline under varying data scales on real robot. The x-axis represents the number of expert action samples per task used during training, and the y-axis shows the corresponding success rate (%). It can be observed that ROSA consistently outperforms the baseline in different data scales.

Table 1: Comparison of performance (success rate %) under varying data scales on 12 RLBench tasks.

| Method | 50 | 100 | 200 | 400 | 500 |
|---|---|---|---|---|---|
| Baseline | 18.6 | 52.3 | 78.8 | 86.1 | 90.5 |
| ROSA | 25.7(+7.1) | 63.7(+11.4) | 80.1(+1.3) | 88.0(+1.9) | 92.1(+1.6) |

Table 2: One-shot performance (success rate %) comparison on 3 RLBench tasks.

| Method | Turn Tap | Put Money in Safe | Push Buttons |
|---|---|---|---|
| Baseline | 0 | 0 | 0 |
| ROSA | 4 | 4 | 32 |

SR. This result further highlights the effectiveness of ROSA, demonstrating its benefits even in high-data regimes.

**One-Shot Scenarios.** Another important question is whether ROSA remains effective under extremely low-data conditions. To investigate this, we design a one-shot experiment in which only a single expert action sample is provided during training. Remarkably, as shown in Tab. 2, ROSA achieves non-zero success rates on three tasks, whereas the baseline model fails on all of them. This experiment demonstrates ROSA's strong data efficiency and its ability to enhance the VLA model's understanding of 3D spatial structures and action semantics, even in highly data-scarce settings.

### 4.2.2 GENERALIZATION ABILITY OF ROSA

We evaluate the generalization ability of ROSA using four real-world tasks involving unseen objects, unseen containers, and the presence of distractors. As shown in Tab. 3, ROSA significantly outperforms the baseline across all four tasks in terms of success rate. While the baseline benefits from the prior knowledge provided by VLM pretraining and exhibits some generalization ability—for example, achieving a 50% success rate in grasping the unseen cube—it performs worse compared to seen objects such as bananas or strawberries. The performance drop becomes even more pronounced in novel scenarios involving both unseen objects and containers, where the success rate falls to around 20%. In contrast, ROSA consistently demonstrates strong generalization performance. Notably, on the *Cube in Box* task, ROSA exceeds the baseline by 60%. Furthermore, ROSA achieves a 90% success rate on Strawberry in Bowl (with distractors), which is 30% higher than the baseline, highlighting its robustness to distracting objects.

### 4.2.3 COMPARISON WITH PREVIOUS METHODS

We compare ROSA with previous methods. As shown in Tab. 4, on RLBench, compared to the VLA-based method LLARVA, ROSA achieves a 16-point improvement in success rate, despite LLARVA using 800 expert demonstrations per task—eight times more data than ROSA. This result highlights both the strong performance and high data efficiency of ROSA. Additionally, compared to the non-VLA method PerACT, ROSA achieves higher performance, even though PerACT utilizes multiple cameras and depth information. On real robots, as shown in Tab 5, we compare our method with state-of-the-art VLA approaches, $\pi 0$ and CogAct. It is evident that ROSA demonstrates a significant performance advantage—for instance, achieving a 33% gain over $\pi 0$. This highlights ROSA's superior effectiveness in low-data real-world robotic settings.

Table 3: Performance on unseen tasks for real robot.

| Method | Cube in Plate | Strawberry in Box | Cube in Box | Strawberry in Bowl(dist) | Average SR |
|---|---|---|---|---|---|
| Baseline | 50% | 40% | 20% | 60% | 43% |
| ROSA | 90% | 80% | 80% | 90% | 85% |

Table 4: Comparison with previous methods on RLBench. We compare the success rate (%) on 12 different tasks. ROSA shows superior performance compared with these related methods.

| Method | Method Type | Average SR | Open drawer | Slide block | Sweep dustpan | Meat off grill | Turn tap |
|---|---|---|---|---|---|---|---|
| Image-BC (CNN) (Jang et al., 2022) | non-VLA | 2.3 | 4 | 4 | 0 | 0 | 8 |
| Image-BC (ViT) (Jang et al., 2022) | non-VLA | 1.3 | 0 | 0 | 0 | 0 | 16 |
| C2FARM-BC (James et al., 2022) | non-VLA | 22.3 | 20 | 16 | 0 | 20 | 68 |
| PerAct (Shridhar et al., 2023) | non-VLA | 57.3 | 80 | 72 | 56 | 84 | 80 |
| LLARVA(Niu et al.) | VLA | 47.7 | 60 | 100 | 84 | 80 | 56 |
| ROSA | VLA | 63.7 ± 0.6 | 66.7±4.8 | 74.7±4.8 | 82.7±4.8 | 65.3±1.3 | 66.7±5.3 |

| Method | Put item | Close jar | Reach and drag | Put money | Place wine | Push buttons | Put groceries |
|---|---|---|---|---|---|---|---|
| Image-BC (CNN) (Jang et al., 2022) | 8 | 0 | 0 | 4 | 0 | 0 | 0 |
| Image-BC (ViT) (Jang et al., 2022) | 0 | 0 | 0 | 0 | 0 | 0 | 0 |
| C2FARM-BC (James et al., 2022) | 4 | 24 | 24 | 12 | 8 | 72 | 0 |
| PerAct (Shridhar et al., 2023) | 68 | 60 | 68 | 44 | 12 | 48 | 16 |
| LLARVA (Niu et al.) | 0 | 28 | 52 | 44 | 12 | 56 | 0 |
| ROSA | 70.7 ± 1.3 | 73.3 ± 7.1 | 77.3±1.3 | 60.0 ±4.6 | 46.7 ± 1.3 | 65.3 ±4.8 | 8.0 ±2.3 |

Table 5: Comparison with previous methods on real robot.

| Method | Banana in plate | Strawberry in Bowl | Starfruit in Plate | Banana&Strawberry in Bowl | Average SR |
|---|---|---|---|---|---|
| $\pi_0$ (Black et al., 2024) | 50% | 30% | 40% | 20% | 35% |
| CogACT (Li et al., 2024b) | 40% | 30% | 30% | 20% | 30% |
| ROSA | 90% | 70% | 70% | 40% | 68% |

### 4.2.4 GENERALITY OF ROBOT STATE ESTIMATION

To verify the universality of robot state estimation for ROSA, we incorporate robot state data into the training of another VLA model, CogACT, and measure the resulting performance improvements. As shown in Tab. 6, adding robot state data significantly boosts the performance of vanilla CogACT, demonstrating that ROSA can be seamlessly applied to different VLA frameworks and bring consistent performance gains.

Table 6: Impact of incorporating ROSA into CogACT on real-robot performance.

| Method | Banana in plate | Strawberry in Bowl | Starfruit in Plate | Banana&Strawberry in Bowl | Average SR |
|---|---|---|---|---|---|
| CogACT (Li et al., 2024b) | 40% | 30% | 30% | 20% | 30% |
| CogACT + State Estimation | 60% | 70% | 80% | 50% | 65% |

### 4.2.5 ANALYSIS OF ROSA

To better understand how ROSA works, we conducted controlled studies in the RLBench simulation environment. All experiments used 100 expert action samples per task.

**How much robot state data is needed?** We investigate the effect of incorporating different amounts of robot state data, as shown in Tab. 7. Starting with expert action data only, adding just one-eighth of the state data yields a 3.7% improvement in success rate. Increasing the proportion to one-quarter leads to a further 7.7% gain. Using larger amounts of state data degrades performance, likely due to distributional shifts that impair the model's ability to predict future actions. These results suggest that incorporating a relatively small amount of robot state data is sufficient to yield substantial performance improvements, indicating the effectiveness of introducing robot state data.

**How should the environment be configured for state data collection?** We study the impact of scene type and scene quantity. Specifically, we consider two types of scenes: (1) *relevant scenes*, which share the same setup as the evaluation tasks, and (2) *irrelevant scenes*, which feature unrelated configurations without the evaluation subjects. Fig. 5 (b) provides visualizations of these two scene types. Scene quantity refers to the number of distinct spatial arrangements within a given scene setup.

Tab. 8 presents the results of our analysis. We find that scene relevance is not critical—both relevant and irrelevant scenes lead to comparable improvements in performance. This finding suggests that, in practice, environments originally used for collecting expert action data can be effectively reused for robot state data collection, avoiding the need to design new scenes. Additionally, we observe that a moderate scene quantity (e.g., 100 distinct scenes) is sufficient to achieve optimal performance.

**3D understanding capability of ROSA.** To evaluate whether robot state data truly enhances the alignment between a pre-trained VLM and robot-specific representations, we conduct a linear probing analysis. Specifically, we add a linear layer on top of the LLM and train it on a newly constructed 3D spatial understanding dataset. The task requires the model to predict the 3D position and orien-

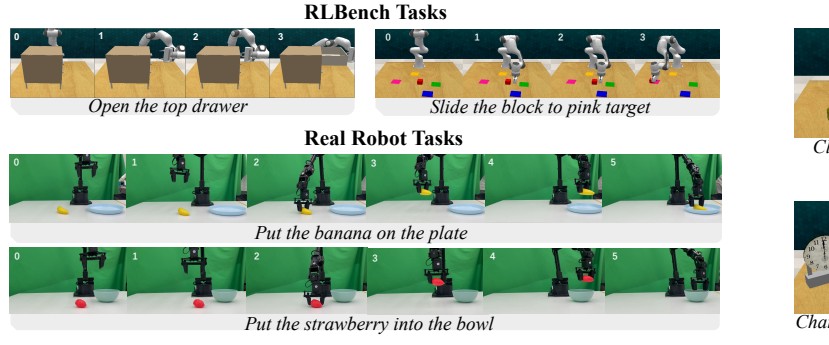

(a) Visual examples on RLBench and real-world robot tasks    (b) Examples of robot state data

Figure 5: Visualization results. (a). Examples of ROSA on RLBench and real-world robot tasks. (b). Examples of robot state data for relevant secenes and irrelevant scenes

Table 7: Ablation on the data ratio.

| State/Action Ratio | SR |
|---|---|
| 0 | 52.3 |
| 1/8 | 56.0 |
| 1/4 | 63.7 |
| 1/2 | 58.7 |

Table 8: Ablation on scene types and quantity of robot state data.

| Scene Type | # Scenes | SR |
|---|---|---|
| Relevant | 50 | 58.1 |
| Relevant | 100 | 63.7 |
| Relevant | 200 | 62.9 |
| Irrelevant | 100 | 63.7 |

Table 9: Liner-prob evaluation on 3D understanding.

| Method | Acc. % | Error |
|---|---|---|
| VLM | 0 | 5.2 |
| Baseline | 61 | 2.1 |
| ROSA | 92 | 1.4 |

tation of the robot's end-effector based on a single image. We compare three models: the pre-trained VLM, a baseline VLA model trained only with expert action data, and ROSA. We use mean squared error (MSE) and prediction accuracy under a fixed error threshold as evaluation metrics.

As shown in Tab. 9, the pre-trained VLM achieves 0% accuracy, indicating a lack of 3D spatial understanding. The baseline VLA model attains 61% accuracy, suggesting some capacity to perceive 3D information. ROSA achieves the highest accuracy of 92%, demonstrating significantly improved 3D spatial reasoning. These results indicate that ROSA effectively bridges the spatial representation gap between VLMs and robot-specific learning.

### 4.3 QUALITATIVE RESULTS

**Visualizations of Robot State Data:**   The robot state data is collected by allowing the robotic arm to perform random movements within its valid action space. We set up different scenarios for collecting such data and the examples are illustrated in Fig. 5 (b) . For relevant scenes, the robot state data is collected in the same scenes as evaluation data but the movements are random. For irrelevant scenes, we choose completely different task settings from RLbench.

**Examples of ROSA on Simulation and Real Robot Tasks:**   Fig. 5 (a) illustrates ROSA's performance on several tasks in both simulation and real-world environments. The top roll in Fig. 5 (a) presents results on two RLBench tasks: *Open Drawer* and *Slide Block to Color Target*. It can be observed that the model accurately predicts the actions and effectively manipulates the target objects. Real-robot task executions are also shown in Fig. 5 (a), where ROSA precisely localizes the target objects and their corresponding containers, and successfully place the objects into the containers.

### 5 CONCLUSION

We propose ROSA, a novel framework that leverages robot state estimation data to better align vision-language and action spaces in VLA models. By improving the model's understanding of its own 3D state, ROSA enables more accurate future action prediction. We also introduce an automated data collection method that requires nearly no human efforts. Extensive experiments in both simulation and real-world settings demonstrate ROSA's effectiveness, showing strong performance and generalization ability. In future work, ROSA can be extended to a wider range of tasks and architectures, even applied to cross-platform or cross-robot scenarios.

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

## APPENDIX

In the appendix, we provide more experimental details and qualitative visualizations of ROSA. We also discuss the current limitations and outline promising directions for future work.

## A    MORE IMPLEMENTATION DETAILS

### A.1    MODEL AND TRAINING DETAILS

We build ROSA based on the Qwen-2.5-7B (Yang et al., 2024) model as the LLM backbone, CLIP ViT-L/14 (Radford et al., 2021) as the vision encoder, and a two-layer MLP as the projector. In our initial experiments, we observed that using a stronger language model led to better overall performance on VLA tasks. We compared two strong LLMs—Qwen2.5-7B and LLaMA3-8B[10]—and found that Qwen2.5 achieved a 6% relative improvement after finetuning, while also having fewer parameters. Based on these results, we selected Qwen2.5 as our base LLM, as its strong capability in domain-specific applications offers a solid starting point for VLA models. Ongoing advances in LLMs and VLMs are expected to further enhance ROSA's overall performance. ROSA mix the robot state data and the expert action data in a fixed ratio of 1:4 and performs joint training. We fully fine-tune all layers for 6 epoches on RLbench and 9 epoches on real robot. A learning rate of 2e-5 is adopted with a warmup and cosine-decay scheduler. All experiments are conducted on 8 NVIDIA A100 GPUs.

### A.2    RLBENCH SETUPS

RLBench is a widely used simulation benchmark for robotic manipulation, encompassing a diverse set of tasks with varying levels of difficulty. In our experiments, we select 12 representative tasks to evaluate the multi-task performance of ROSA. Each task contains multiple varia- tions during data collection but remains consistent during evaluation to enable one-to-one compar- isons. For training, we follow the data generation pipeline from PerAct (Shridhar et al., 2023) to create demonstrations for each task. Detailed descriptions of the selected tasks are provided below.

**Open Drawer:** The task involves a drawer with upper, middle, and lower compartments. The task is to pull out the correct drawer specified by the instruction beyond a set distance.

**Slide Block:** The scene includes four colored square stickers and a cube block. The task is to slide the block to the target region of correct color.

**Sweep Dustpan:** The scene contains one large and one small dustpan, with debris scattered between them. The task is to grab the broom, sweep it at an angle, and push all debris into the specified dustpan. Success is defined as the complete removal of all debris in the correct dustpan.

**Meat off Grill:** The scene contains two types of meat on the grill: steak and chicken leg. The task is to pick up the specified meat and place it on the metal rack next to the oven.

**Turn Tap:** The objective of this task is to rotate either the left or right handle of a faucet, with the left and right sides determined by the faucet's orientation.

**Put Item:** The drawer has upper, middle , and lower compartments, with a small block placed on top. The task is to pick up the block and place it into the target drawer.

**Close Jar:** The task scenario includes two jars with different colors and a lid. The task is considered successful if the lid is placed on the correct jar and the robot gripper is empty.

**Reach and Drag:** The scene contains a rod-shaped tool, a movable block, and four colored target zones. The task is to grasp the rod and push the block into the correct colored area.

**Put Money:** The scene features a three-layer safe with a bundle of US dollar bills on top. The task is to grasp the bill and place it accurately in the specified compartment of the safe.

**Place Wine:** The scene includes a red wine bottle and a wine rack. The task is to grasp the bottle by the neck and securely place it in the correct position on the rack.

**Push Buttons:** The scene involves multiple buttons of different colors. An example instruction is "push the maroon button, then push the green button". The task is to push buttons of different colors in the order specified by the instruction.

**Put Groceries:** The scene contains nine grocery items and one cupboard. The task is to correctly place the item specified by the language instruction into the cupboard.

### A.3  REAL ROBOT SETUPS

**Hardware Setup:**  We conduct real-robot experiments using a Trossen WidowX 250s 6-DOF robotic arm. We use an Intel RealSense D435i camera mounted on a tripod at a fixed angle to capture visual observations including both the manipulator and objects on the tabletop, as shown in Fig. 6. The camera provides images at a resolution of 1280×720 and a frame rate of 30 Hz.

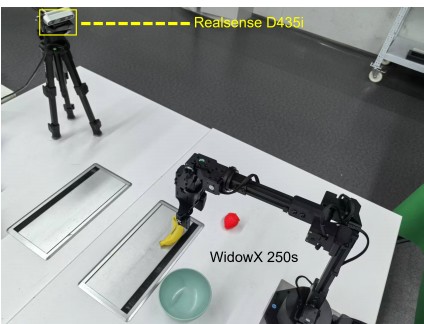

Figure 6: Real-robot setup with WidowX 250s and D435i.

**Data Collection:**  We developed a custom system for data collection. A human operator controls the robotic arm using a gamepad to perform various tasks. Meanwhile, a camera records real-time images, and the data collection program logs the real-time pose of the end-effector and the gripper state. For the pose, we record the end-effector's position and orientation relative to the robot base coordinate system; for the gripper state, we record the distance between the two fingers of the gripper. During data collection, the operator randomly places objects within a reasonable range to gather diverse data. To improve training stability, we additionally collected a set of auxiliary tasks that differ from the evaluation tasks. These tasks serve as supplemental training data—for example, "Put Eggplant in Plate".

### A.3.1 SEEN TASKS

We evaluated ROSA on four seen tasks which are included during training. The details of these tasks are as follows.

**Banana in Plate:** The scene contains a banana and a plastic plate on the table. The task is to pick up the banana and place it onto the plate.

**Strawberry in Bowl:** The scene has a strawberry and a bowl on the table. The objective of this task is to grasp the strawberry and place it safely into the bowl.

**Starfruit in Plate:** The scene contains a starfruit and a plate on the table. The task involves picking up the starfruit and placing it on the plate.

**Banana and Strawberry in Bowl:** The scene contains a strawberry, a banana and a bowl on the table. The task is considered successful if both the banana and the strawberry are successfully placed into the bowl, regardless of the order.

### A.3.2 UNSEEN TASKS

To evaluate the generalization and robustness of the model under novel disturbances, we introduced a set of "unseen tasks" that are never encountered during training. In addition to measuring success rates on four representative unseen tasks, we further extended the evaluation to a broader range of scenarios involving novel objects, containers, and environmental variations, as shown in Sec. B. These experiments demonstrate the strong generalization ability and robustness of ROSA.

**Cube in Plate:** This scene contains a cube and a plate on the table. The task is to pick up the cube and put it on the plate.

**Cube in Box:** This scene contains four cubes with different colors including blue, red, yellow and green and a paper box. The target of the task is to pick the red cube into the paper box. The task is considered successful if the robot arm selects the cube of the correct color and successfully places it into the paper box.

**Strawberry in Box:** The scene has a strawberry and a paper box. The task is to pick up the strawberry and put it to the paper box.

**Marker in Plate:** This scene contains a marker pen and a plate on the table. The target of this task is to put the marker pen on the plate.

**Put Banana in Colored Region:** This scene contains four colored regions and a banana on the table. The target of this task is to put the banana in the region of the correct color.

**Put Cube in Cup:** This scene contains four cubes of different colors and a blue cup on the table. The target of this task is to choose the cube of the correct color and put it into the blue cup.

**Put Multiple Cubes in Plate:** The scene has two or more cubes on the table, the target of this task is to put all the cubes on the plate.

**Strawberry in Bowl (dist):** The scene has a strawberry, a bowl, and many different fruits or vegetables as distractors, such as grapes and corn. The objective of this task is to place the strawberry into the bowl in the presence of multiple distractor objects. The task is considered successful if the strawberry is placed inside the bowl without picking up any distractors.

**Banana and Strawberry in Bowl (dist):** The scene contains a banana, strawberry, a bowl, and other different fruits or vegetables as distractors. The objective of this task is to place the banana and strawberry into the bowl no matter of the order without picking up any distractors.

### A.4 KEYFRAME EXTRACTION

For both RLBench and real robot experiments, we follow PerAct(Shridhar et al., 2023) to extract keyframes and their corresponding key actions based on the end-effector pose, gripper state, and joint velocity. Specifically, an action is considered a keyframe if (1) the joint velocities are close to zero and (2) the gripper's open state changes abruptly. We estimate the velocity of the robotic arm by

computing the change in the end-effector's position between consecutive frames and always include the first and last frames as keyframes. The detailed process is illustrated in the pseudo code 1.

---

**Algorithm 1** Keyframe Extraction

---

**Input:** Trajectory $D = \{d_0, d_1, \ldots, d_n\}$, where each $d_i$ has position $p_i$ and gripper state $g_i$
**Output:** Keyframe set $K$

$K \leftarrow \{d_0, d_n\}$                                    ▷ Include first and last frames
Initialize recent_buffer as empty list
**for** $i = 1$ to $n - 1$ **do**
    $\Delta p \leftarrow \|p_i - p_{i-1}\| + \|p_{i+1} - p_i\|$
    **if** $\Delta p < \epsilon$ and no static frame in recent_buffer **then**
        $K \leftarrow K \cup \{d_i\}$
        Append $d_i$ to recent_buffer (keep size $\leq 4$)
    **end if**
    **if** $g_i \neq g_{i-1}$ or $g_i \neq g_{i+1}$ **then**
        $K \leftarrow K \cup \{d_i\}$
    **end if**
**end for**
**return** $K$

---

Seen Task

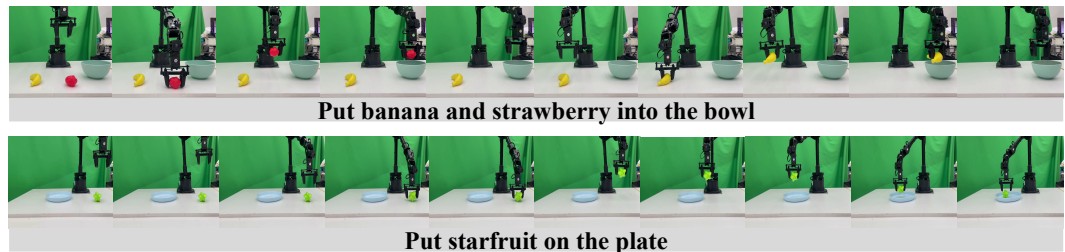

Figure 7: Visual examples of ROSA on seen tasks on real-world robot.

# B    MORE VISUALIZATIONS

We qualitatively showcase ROSA's diverse capabilities through more visualizations. We use a hand-held third-party camera to record the demos during evaluation and provide a video in MP4 format, named "demo.mp4", which is attached to this file. We also provide video frames in this appendix. As shown in Fig. 7, ROSA performs well on the tasks it has been trained on. More importantly, it demonstrates strong generalization and robustness on unseen tasks, as illustrated in Fig. 8. ROSA can successfully manipulate novel objects it has never encountered during training—for example, it can distinguish and grasp cubes specified by color, and it can also handle unfamiliar objects such as a marker pen. Furthermore, ROSA generalizes to unseen containers, such as paper box and even more abstract targets like colored regions. Remarkably, even when both the object and the target container are entirely novel, ROSA can still accomplish the task successfully. In addition, ROSA is capable of handling long-horizon unseen tasks, such as sequentially placing three cubes into a plate as shown in Fig. 9. Beyond object-level generalization, ROSA also shows robustness to distractors. As depicted in Fig. 10, even in cluttered scenes filled with unseen distracting items like various fruits and vegetables, ROSA is able to identify and manipulate the correct target objects and place them into the appropriate container. These visualizations collectively demonstrate that ROSA acts as a powerful generalist policy.

(a) Unseen Task – Unseen objects

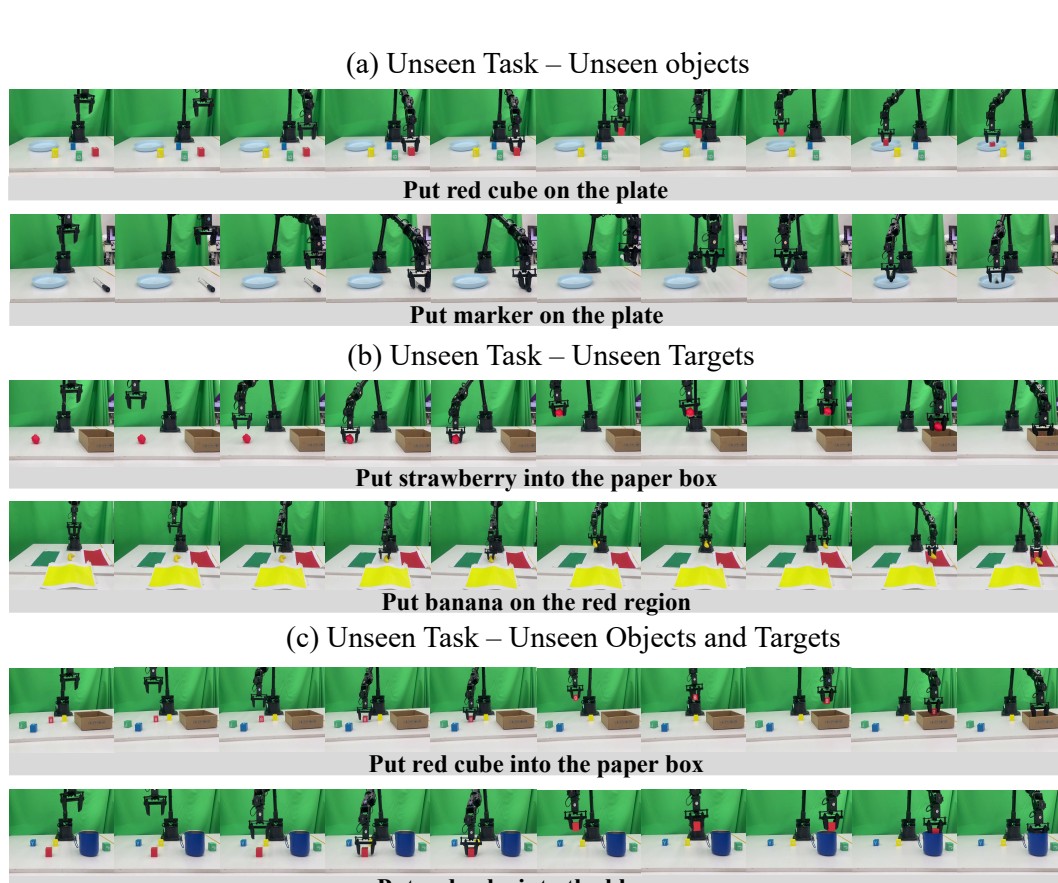

Figure 8: Visual examples of ROSA on unseen tasks on real-world robot.

Long-horizon Unseen Task

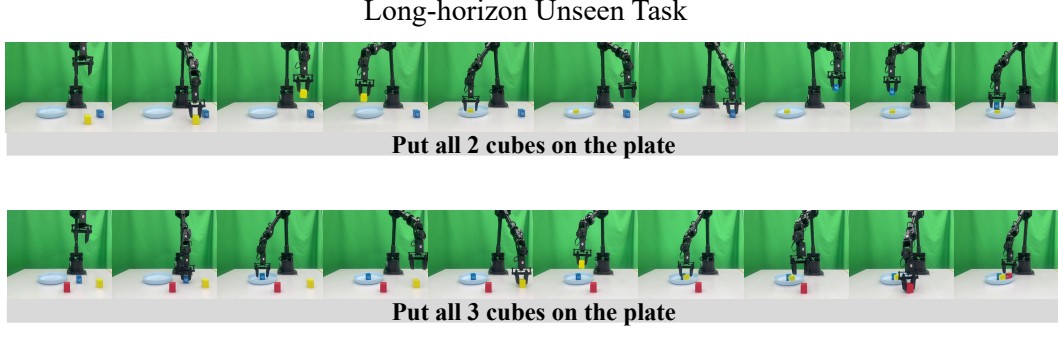

Figure 9: Visual examples of ROSA on long-horizon unseen tasks on real-world robot.

Interference Task

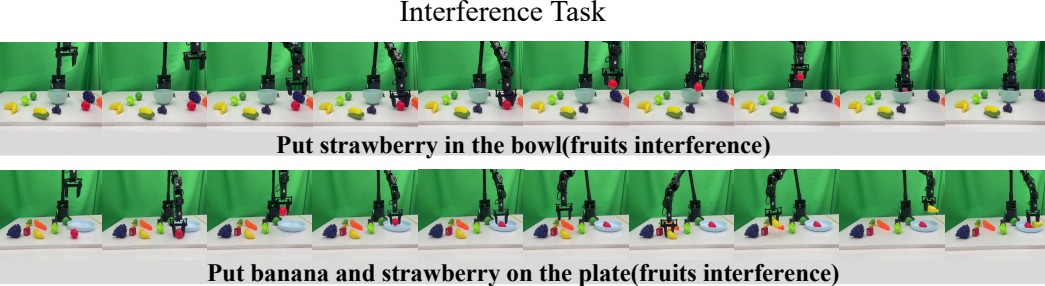

Figure 10: Visual examples of ROSA on interference tasks on real-world robot.

## C    LIMITATIONS

While ROSA provides a scalable and data-efficient training paradigm with minimal reliance on additional human labor, certain limitations remain. Despite surpassing existing VLA methods, a noticeable performance gap persists in low-data regimes when compared with 3D-based non-VLA approaches (Goyal et al., 2023; Ke et al., 2024), reflecting a broader common challenge faced by current VLA models. Incorporating multi-view or depth information offers a promising direction to narrow this gap. Moreover, our current setup assumes a fixed relative pose between the camera and the robot; extending ROSA to more dynamic settings, such as in cross-camera or mobile-camera deployments, represents another compelling avenue for future research.

## D    LLM USAGE

We used LLM solely for checking grammar and polishing the writing.

