# OpenReview forum: "ROSA: Harnessing Robot States for Vision-Language and Action Alignment"
_ICLR.cc/2026/Conference — ICLR 2026 Conference Withdrawn Submission_

### Official Review · Reviewer_THEy · 2025-10-25

**Soundness:** 3
**Presentation:** 3
**Contribution:** 2
**Rating:** 4
**Confidence:** 3

**Summary:**

To mitigate the spatio-temporal gap between VLA and VLM, this paper propose ROSA, integrating robot state estimation as an auxiliary task, to grain spatial understanding and self-awareness, then improve the VLA performance. Evaluation on both simulation and real-world demonstrate the effectiveness.

**Strengths:**

* The method effectively addresses the spatio-temporal misalignment between VLMs and VLAs through a novel integration of robot state estimation.
* The auxiliary task design enhances the model’s spatial reasoning and action prediction capabilities without requiring extensive human annotation.
* Extensive experiments across simulation and real-world environments validate the robustness and generalization ability of ROSA, particularly in low-data regimes.

**Weaknesses:**

While ROSA demonstrates compelling advantages in low-data regimes, its scalability and sustained effectiveness in large-scale data scenarios remain unclear. As the volume of expert demonstrations increases, the relative contribution of the robot state estimation task may diminish.

The core weakness is not that ROSA performs worse, but that its unique value proposition appears to weaken. One would expect a truly robust method to maintain a more consistent performance gap, demonstrating that the spatial understanding it provides is complementary and non-redundant, even when expert data is plentiful.

**Questions:**

Please refer to the "Weaknesses" section.

---

### Official Review · Reviewer_7eKi · 2025-10-27

**Soundness:** 3
**Presentation:** 3
**Contribution:** 2
**Rating:** 4
**Confidence:** 4

**Summary:**

This paper introduces ROSA (RObot State estimation for vision-language and action Alignment), a novel training paradigm designed to improve the data efficiency and performance of Vision-Language-Action (VLA) models. The authors identify a fundamental "spatio-temporal gap" that arises when fine-tuning pretrained Vision-Language Models (VLMs) for robotic control: VLMs understand high-level 2D semantics of the present, whereas robotics requires low-level 3D spatial reasoning to predict future actions. To bridge this gap, ROSA proposes jointly training the VLA model on two data types: (1) standard, human-provided expert demonstrations for future action prediction, and (2) a new type of "robot state estimation data" for current state prediction. This second data type, which pairs images with the robot's corresponding 7-DoF end-effector pose, is collected via a fully automated process involving random robot movements, thus requiring no additional human labor. By forcing the model to explicitly predict its current 3D state, this auxiliary task enhances spatial awareness and provides a better-grounded context for action prediction. The authors validate ROSA through extensive experiments in both the RLBench simulation and on a real-world WidowX robot, demonstrating significant performance gains, especially in low-data regimes.

**Strengths:**

- The core idea of decomposing the VLM-to-VLA alignment problem into two sub-tasks—predicting the current state and predicting the future action—is a novel contribution. This reframing provides a clear mechanism to directly tackle the identified spatial gap by grounding the model in its own physical embodiment.
- The paper is well-written. The motivation is articulated with great clarity, and the core concept of the spatio-temporal gap is well illustrated.
- The authors validate the general applicability of their proposed data type by successfully integrating it into a different VLA architecture (CogACT) and observing significant performance gains.

**Weaknesses:**

- While the paper strongly justifies how the state estimation task bridges the spatial gap, the argument for bridging the temporal gap is less direct. The auxiliary task focuses on estimating the current state ("What is the current state of the robot?"). While a better understanding of the present undoubtedly aids in predicting the future, the mechanism does not explicitly train the model on temporal dynamics or future forecasting beyond what the standard expert demonstrations already provide.
- The automated data collection relies on "plausible random actions" within a constrained space. While the ablation showing that scene relevance is not critical is an interesting and practical finding, the strategy of uniform sampling within the workspace might be suboptimal. It's plausible that this uniform distribution underrepresents regions of the state space that are more complex or critical for specific tasks.
- The primary baseline for ROSA is a custom model built on Qwen-2.5-7B. While strong, demonstrating the method's effectiveness by integrating it into a more widely adopted, state-of-the-art open-source VLA like OpenVLA or $\pi_0$ as the primary baseline would have further strengthened the claims of broad applicability.

**Questions:**

Could you elaborate on the mechanism by which ROSA helps bridge the temporal gap? Is the improvement in future action prediction solely a consequence of better spatial grounding in the present, or is there a more direct temporal reasoning component being learned that I may have missed?

---

### Official Review · Reviewer_oZ6R · 2025-10-28

**Soundness:** 2
**Presentation:** 2
**Contribution:** 1
**Rating:** 2
**Confidence:** 4

**Summary:**

To address the issue that traditional VLA models struggle to align low-dimensional action spaces with high-dimensional vision-language spaces, making it difficult to fully leverage their information, this paper proposes ROSA—a paradigm that predicts robot proprioceptive states using visual-language information and aligns the two spaces. Furthermore, to facilitate the training of ROSA, the paper introduces an automated tool for collecting state prediction data, enabling rapid adaptation of the ROSA method to various VLA models.

**Strengths:**

1.	The proposed ROSA demonstrates data efficiency. With limited training data, the model can leverage estimated robot proprioceptive states to better understand scene information.
2.	ROSA shows significant improvements over baseline models across multiple RLBench simulation environments and real-world tasks, highlighting its enhanced 3D spatial perception capability focused on robot state.

**Weaknesses:**

1.	The paper only introduces a method for enhancing robot proprioceptive state prediction without delving into how this enhancement affects model representation and reasoning capabilities. The proposed automated data collection paradigm is relatively simplistic, and the overall contribution and novelty appear limited. The authors could further investigate the impact on model representation abilities.
2.	While experiments are conducted in both simulated and real-world environments, the simulation comparisons only include baseline models without ROSA and several non-VLA models. It would be valuable to include comparisons with models such as $\pi_0$ and CogACT in simulators, as simulated environments offer fairer and more reproducible conditions. I recommend adding such validation.

**Questions:**

The methodology section does not clarify how the model is trained. Is the robot state estimation trained independently from action prediction (i.e., in separate stages), or are they trained jointly (with only differing language instructions)? Clarifying this would strengthen the paper's persuasiveness.

---

### Official Review · Reviewer_CUZU · 2025-11-01

**Soundness:** 3
**Presentation:** 3
**Contribution:** 2
**Rating:** 2
**Confidence:** 4

**Summary:**

The paper proposes ROSA, a training paradigm for Vision-Language-Action (VLA) models that augments standard expert demonstration fine-tuning with an auxiliary, fully automated robot state estimation task. The core idea is to mitigate the spatial and temporal gaps between VLM pretraining (high-level, current-scene semantics) and robotic control (low-level, future actions in 3D) by jointly training on two data types.

**Strengths:**

1. Clear and focused reframing of the VLA alignment problem via explicit robot state estimation as an auxiliary task.
2. Method is well-grounded: the spatial-temporal gap motivation is compelling and supported by analyses (e.g., linear probing).
3. The writing is good and easy-to-follow.

**Weaknesses:**

1. The paper notes that too much state data can degrade action prediction, but the mitigation strategy is limited to a fixed mixing ratio. Deeper analysis on curriculum, sampling strategies, or domain-adaptive weighting could clarify best practices.

2. The setup assumes fixed camera-robot extrinsics. Mobile or cross-camera settings remain open.

3. Evaluation experiments on LIBERO or CALVIN are missing, which are two commonly used VLA evaluation benchmarks.

4. The state estimation as an auxiliary task is good, but only with this is a little weak for a top-ai-conference.

5. Very limited real-world experiments, and currently only focusing on the table-top pick-place tasks. When the robot is conducting some dexterous tasks, the self-occulsion is inevitably, can the ROSA still works?

**Questions:**

1.  quantization errors for rotation?

2. see weakness.

---

### Note · Authors · 2025-11-12

I have read and agree with the venue's withdrawal policy on behalf of myself and my co-authors.